# Historical Tsunamis of Taiwan in the Eighteenth Century: the 1781 Jiateng Harbor Flooding and 1782 Tsunami Event

Tien-Chi Liu[1,2], Tso-Ren Wu[3,4,5], Shu-Kun Hsu[6,7]

1 Center for Astronautical Physics and Engineering, National Central University, Taiwan
5   2 Department of Space Science and Engineering, National Central University, Taiwan
3 Graduate Institute of Hydrological and Oceanic Sciences, National Central University, Taiwan
4 Department of Civil Engineering, National Central University, Taiwan
5 Earthquake-Disaster & Risk Evaluation and Management Center, National Central University, Taiwan
6 Department of Earth Sciences, National Central University, Taiwan
10  7 Institute of Earth Sciences, Academia Sinica, Taipei, Taiwan

*Correspondence to*: Tso-Ren Wu (tsoren@ncu.edu.tw)

**Abstract.** This research aims to study two historical tsunamis occurred in Taiwan during the 18th century and to reconstruct the incidents. The 1781 Jiateng Harbor Flooding, recorded by the Chinese historical document, titled Taiwan Interview Catalogue, took place on the southwest coast of Taiwan. On the other hand, the 1782 Tsunami was documented in foreign languages, with uncertainties about the actual time. These two events seem to be close enough in time and location that, to some researchers, they are considered as the same event. Reasoning these historical events requires carefully examining the literature records and performing the scenarios that match the descriptions. The Impact Intensity Analysis (IIA) is employed to locate possible regions of tsunami sources in order to reproduce the events. Numerical simulations based on the Cornell Multi-Grid Coupled Tsunami Model (COMCOT) analyze the influence of different types of tsunamis generated both by submarine mass failures and seismic activities. Numerical results indicate that the source of the 1781 Jiateng Harbor Flooding is located very possibly on the SSW side of Taiwan. However, simulation results and historical records put the existence of the 1782 Tsunami in doubt, and the possibility of storm surges could not be ruled out.

## 1 Introduction

One of the major hazards in coastal regions is inundation by water waves generated by different mechanisms such as storm surges, tsunamis, or meteotsunamis. Storm surges and meteotsunamis are known to be triggered by weather events associated with pressure changes. On the other hand, tsunamis could be generated by earthquakes below or near the ocean, submarine landslides, volcanic eruptions, meteorite impacts, or off-shore rock falls, causing damage to the infrastructures and large loss of life in the coastal areas (Ghobarah et al., 2006; Mori et al., 2013; Widiyanto et al., 2019). On 15th January 2022, the volcanic eruption of Hunga Tonga-Hunga Ha'apai generated tsunami waves that triggered evacuations in the surrounding countries. The sea-surface fluctuations were widely recorded and raised world-wide attention on the issue of coastal inundations (Carvajal et al.,2022; Manneela and Kumar, 2022; Ramírez-Herrera et al., 2022). To prepare for possible natural

disasters in Macau, Li et al. (2018) have assessed the future tsunami hazard evolving with a rising sea level.   Potential tsunami hazard assessments in the South China Sea (SCS) have also been extensively studied over recent years (Li et al., 2016; Liu et al., 2009; Megawati et al., 2009; Okal et al., 2011).   However, it is still essential to look back and examine the historical events in order to recognize the regions that could be affected again.   In fact, although most historical tsunamis were reviewed already (Lau et al., 2010; Terry et al., 2017), some of the events in Taiwan remain unknown.

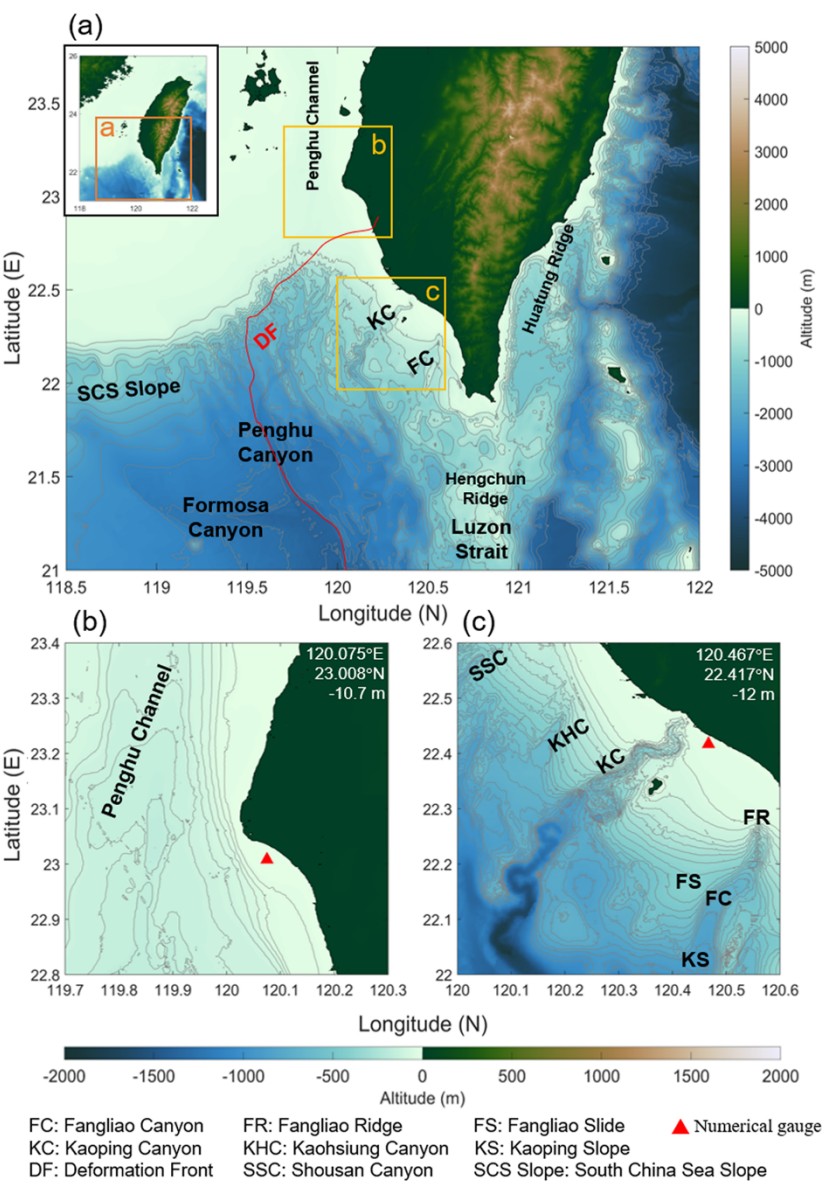

**Figure 1 The area of study.   (a) Bathymetry and related structures off South Taiwan with 250 m contour interval. (b, c) Zoomed maps of Tainan and Pingtung (i.e. Jiateng Harbor location). The contours are plotted with 20 m and 50 m, respectively. Red triangles denote the numerical gauges in this study and the gauge information are marked in the right corner of each figure.**

Located in the typhoon-prone area and the Circum-Pacific Belt, Taiwan suffers from both low-pressure weather systems and seismic activities.   As the historical storm surge events have been studied (Huang et al., 2007; Tsai, 2014), historical tsunami events have also been modeled (Ma and Lee, 1997; Okal et al., 2011), particularly for the 1867 Keelung event, which is the only historical tsunami incident officially recognized by the government (Cheng et al., 2016; Chung, 2018; Lee, 2014).   The 1781 Jiateng Harbor Flooding and 1782 Tsunami event have received attention as well (Li et al., 2015; Mak and Chan, 2007;

Okal et al., 2011).   Nevertheless, these two incidents which seem close in time and location stay mysterious with unfounded tsunami sources (Fig. 1).

Customs and local news of Taiwan during the Qing dynasty were recorded in Taiwan Interview Catalogue (Chen, 1830).   The Chinese historical document indicates a destructive flooding event occurred at Jiateng harbor, submerging the villages located in southwest Taiwan in 1781 (See Text S1 from Li et al., 2015).   At that time, the weather was fine.   Hence, it is less likely

to have storm surges or meteorological tsunamis.   However, the phenomena described in the historical report match well typical features that usually appear in tsunami events (the sea roared like thunder, giant wave appeared, water retreated quickly, leaping fishes and shrimps left on the ground).   Therefore, the 1781 Jiateng Harbor Flooding has been viewed and formally listed as a historical tsunami in previous researches (Hsu, 2007; Lau et al., 2010; Li et al., 2015; Lin, 2006; Mak and Chan, 2007; Yu, 1994).

Since there were no local earthquake movements recorded in the historical report of 1781 Jiateng Harbor Flooding, Yu (1994) considered the long-distance earthquake as a possible source of the tsunami.   Nevertheless, none of the historical earthquakes (Ganse and Nelson, 1982) could be found to fit the location and the time of the 1781 Jiateng Harbor Flooding.   Note that tsunamis might be generated by other far-field activities, such as submarine volcanic eruptions.   Omori (1916) recorded a tsunami caused by the eruption of Sakurajima on the afternoon of 11 April 1781 and during this event, three boats were

overturned and 15 people were drowned.   However, due to the spatial distance and topographic barrier, the tsunami waves should first arrive at Okinawa and the northeast part of Taiwan, and therefore would have less influence on Taiwan's southwest coast.

The records of the 1782 Tsunami event could only be found in foreign documents (Gazette de France, 1783; Jäger, 1784; Mallet, 1854; Perrey, 1862; Soloviev and Go, 1984).   Based on the descriptions of Gazette de France (1783), Okal et al. (2011)

performed an earthquake-induced tsunami scenario in the Taiwan Strait, and found that the tsunami waves would mainly be confined in the strait.   Nevertheless, because the occurring time is close to the 1781 Jiateng Harbor Flooding, the 1782 Tsunami event has been considered as the same event in some of the previous studies (Li et al., 2015; Lin, 2006).   Lin (2006) suspected the 1781 Flooding event as a typographical error of the 1721 Tsunami event, which also occurred on the southwest coast of Taiwan and affected Tainan.   Meanwhile, Lin (2006) regarded the 1782 Tsunami documented in Russian as another

typographical error of the 1867 Keelung Tsunami by evaluating the credibility of the historical records.   Li et al. (2015), on the other hand, eliminated the seismic movements for the lack of historical reports in the neighboring countries and concluded that the severe tsunami of 1781/1782 was most likely generated by submarine mass failures (SMFs).   However, assuming that the 1781 Jiateng Harbor Flooding and the 1782 Tsunami were independent incidents, we found that Gazette de France (1783)

and the German report of Jäger (1784) should also be listed as the first and important sources of information about the 1782
Tsunami event (see Text A1 and A2).   First of all, the publication date of Gazette de France (1783) is the closest to the event's
year.   Second, from the words "Voici ce que je lis dans J. L. Ab Indagine L. M." (i.e. "Here's what I read from J. L. Ab
Indagine L. M.") in Perrey (1862), it is reasonable to believe that most of the content in the document is quoted from the
German report Philosophisch und physikalische Abhandlungen (Jäger, 1784).   After examining these two documents, it is
also suspected that the date of a second letter sent from Beijing to Versailles was reported with a typographical error about the
year.   Indeed, the date "En décembre 1682" is found in Perrey's (1862) record while the phrase is "Im Decemb. Des 1782" in
Jäger's (1784) report, making the former less reliable.

Severe coastal impacts caused by storm surges have been recorded and studied.   The Bay of Bengal was struck by Bhola
cyclone in 1970 and more than 250,000 human lives were taken away resulting from the storm surge and the flooding it
triggered (Hossain et al., 2008).   In 2005, the storm surge associated with Hurricane Katrina damaged tremendously the Gulf
of Mexico, peaking at over seven meters along the Mississippi coastline (Fritz et al., 2007; Robertson et al., 2007).   The high-
water marks of Cyclone Nargis surpassed the 2004 Indian Ocean Tsunami run-up height at corresponding locations in May
2008 (Fritz et al., 2009), causing over 100,000 deaths in Myanmar.   In southwest Taiwan, a unique local ritual (i.e.
Quianshuizang) is still being practiced today to pacify the victims of the 1845 Yunlin Kouhu Storm Surge Event (Chang, 2019).
The storm surge, inducing by a super-typhoon, was responsible for the fatalities of over 2,000 (Tsai, 2014; Yin, 2013), and
was followed by famine and plague (Chang, 2019).   In the year 1782, Taiwan was hit by a violent storm that led to coastal
damage and vessels missing.   Historical records written in both Chinese and foreign languages were found for this incident
(Davidson, 1903; Griffiths, 1785; Grosier, 1787; Hsu, 2007; Rees, 1820; Yin, 2017).   According to the records of Griffiths
(1785) and Grosier (1787), a typhoon struck Taiwan on 22nd May, which is the same date documented in the 1782 Tsunami
reports (Gazette de France, 1783; Jäger, 1784; Mallet, 1854; Perrey, 1862; Soloviev and Go, 1984).   Other similar descriptions
such as the long-lasting time of natural hazard, impact area (Tainan), coastal inundation, and the imperial edict should also be
highlighted.

In this paper, we assess the tsunami hazard of southwest Taiwan by a new analysis method and conducted numerical
simulations to study the 1781 Jiateng Harbor Flooding and the 1782 Tsunami Event.   The Impact Intensity Analysis (IIA)
method and the modeling approaches of the tsunami scenarios are introduced in Sect. 2.   Numerical simulation results are
analyzed in Sect. 3 and the discussions through the literature records and the model results are presented in Sect. 4.

## 2 Methods

### 2.1 Impact Intensity Analysis

In order to investigate and reconstruct the 1781 Jiateng Harbor Flooding and the 1782 Tsunami, we need to find out the possible

source of the two events, respectively.   To do so, the Impact Intensity Analysis (IIA) was applied to quantify the effect of

potential tsunamis and eliminate the source locations at which the generated tsunami waves are less influential to the study

sites.   This method has been adopted to reconstruct the 1867 Keelung tsunami event (Chung, 2018; Lee, 2014) as well as to

analyze the potential tsunami threats in Taiwan (Wu, 2017).

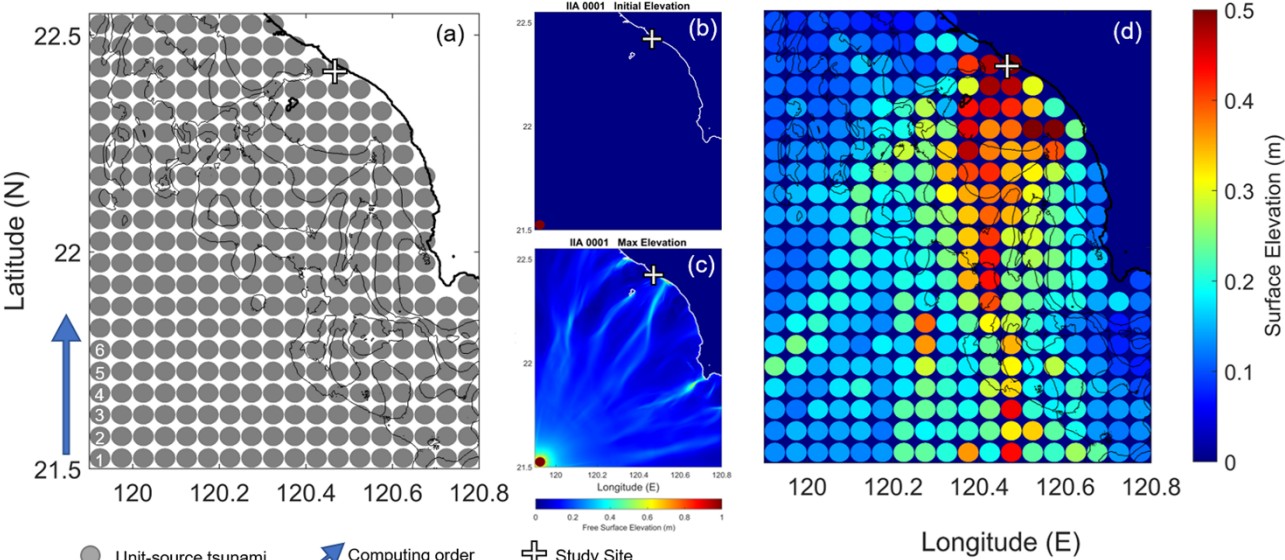

**Figure 2 IIA method procedure. (a) Unit-source tsunamis are set in the discretized domain of the numerical simulation and conducted individually. (b, c) The results of the first unit-source tsunami are presented here as an example. The initial elevation (b) and the maximum wave height (c) are the essential data that should be saved for each simulation. (d) Each simulation's source location is colored by the maximum wave height recorded at the study site (marked by the white cross). Together they form a map comparing the impact of tsunamis in the study area.**

Unit-source tsunamis have been used for tsunami early warning and forecasting studies (Greenslade and Titov, 2008; Horspool

et al., 2014; Liu et al., 2009; Matias et al., 2013; Percival et al., 2011).   Since the IIA method aims to compare the effect of

same-size tsunamis generated at different locations, the shape of unit sources was therefore set up as a cylinder to ensure the

isotropy of initial elevation.   With the unit sources distributed evenly, the analysis method enables us to examine thoroughly

the study areas.   The IIA method is performed with the Cornell Multigrid Coupled Tsunami Model (COMCOT) (Liu et al.,

1998; Wang and Power, 2011) and the process is presented with an example in Fig.2.   As unit-source tsunamis simulations

being conducted individually in the discretized region (Fig. 2a), we preserved only the initial elevation and the maximum wave

height (Fig.2b, 2c) for computational efficiency.   The maximum wave height at the assigned location would be extracted from

each numerical scenario and colored at its source location, forming a map which compares the impact of tsunamis to the study site in the specific area (Fig. 2d).

Referring to the Qianlong Taiwan Map which was drawn between 1756-1759, the study site of Jiateng Harbor is assigned at 120.467°E, 22.417°N, at today's Donggang Township of Pingtung County, while the Tainan study site is located at 120.075°E, 23.008°N (Fig. 1b, 1c). Model domains cover the southwestern coast of Taiwan (117.80° E-120.00° E, 18.70° N-23.40° N), the near-field area (119.90° E-120.80° E, 21.50° N-22.55° N) as well as the far zone (117.65° E-121.75° E, 15.35° N-23.35° N), with resolutions of 0.5 arcmins, 0.05 arcmin, and 1 arcmin, respectively (Table 1). To ensure the accuracy of bathymetry

settings, the bathymetric data with a resolution of 100 m which were gained fundamentally through compilations of data from different marine cruises (Hsu et al., 1996; Hsu et al., 1998), were adopted for the first two models. On the other hand, the far-field analysis was conducted with 1 arcmin bathymetric data of ETOPO1 from National Oceanic and Atmospheric Administration (NOAA).


**Table 1 Parameters of the unit sources adopted in the IIA method for the three domains.**

| | Southwestern coast | Near-field | Far-field |
|---|---|---|---|
| | 117.80° E-120.00° E<br>18.70° N-23.40° N | 119.90° E-120.80° E<br>21.50° N-22.55° N | 117.65° E-121.75° E<br>15.35° N-23.35° N |
| Diameter (km) | 10 | 5 | 40 |
| Elevation (m) | 10 | 1 | 1 |
| Total number | 1504 | 378 | 200 |

By the IIA result of Jiateng Harbor, tsunamis generated in the offshore region had stronger impacts to the study sites on the

southwestern coast of Taiwan (Fig. 3). The areas where submarine structures are located, including the Fangliao Canyon, Kaohsiung Canyon, Kaoping Canyon, Shousan Canyon, and the upper part of Penghu Canyon should be highlighted. With source locations placed from the study site all the way to the Formosa Canyon, tsunami waves had significant impacts on the ancient location of Jiateng Harbor, whereas for Tainan, the possible source locations were limited. Moreover, tsunamis with the source located at the northern Penghu Canyon and in specific parts of the Penghu channel and South China Sea Shelf lead

to greater impact to the study sites in Fig. 3. However, tsunamis triggered in the overlap-highlight area of the Penghu Canyon implied causing effects to both the study sites.


From the result of near-field analysis, it is clear that the study site is mostly affected by the tsunamis generated nearby (Fig. 4). The area from the Jiateng Harbor study site to Liuqiu island is highlighted. Note that the maximum wave heights

recorded at the Jiateng Harbor decrease rapidly with the distance. Similar with the result of southwestern coast analysis, tsunamis with source located at the South China Sea Slope, Kaoping Canyon and Kaoping Slope would have greater influence to the Jiateng Harbor study site. Nevertheless, with a broader region in Fig. 5a, the IIA method revealed that tsunamis generated in the Luzon Strait would also be influential to the Jiateng Harbor. Hence, a dramatic effect of tsunamis triggered by the Manila Trench could be expected at the study site. The far-field IIA of Tainan showed patterns that look like the local

result as well in Fig. 5b. With expanded study area, not only the tsunamis that take place in the upper Penghu Canyon and the Kaoping Slope should be taken into consideration, but also the ones that occur in the Luzon Strait. Meanwhile, since tsunamis triggered along the Henchun and Huatung Ridges should contribute a larger wave height to the study site of Jiateng Harbor, we also pay attention to this region.

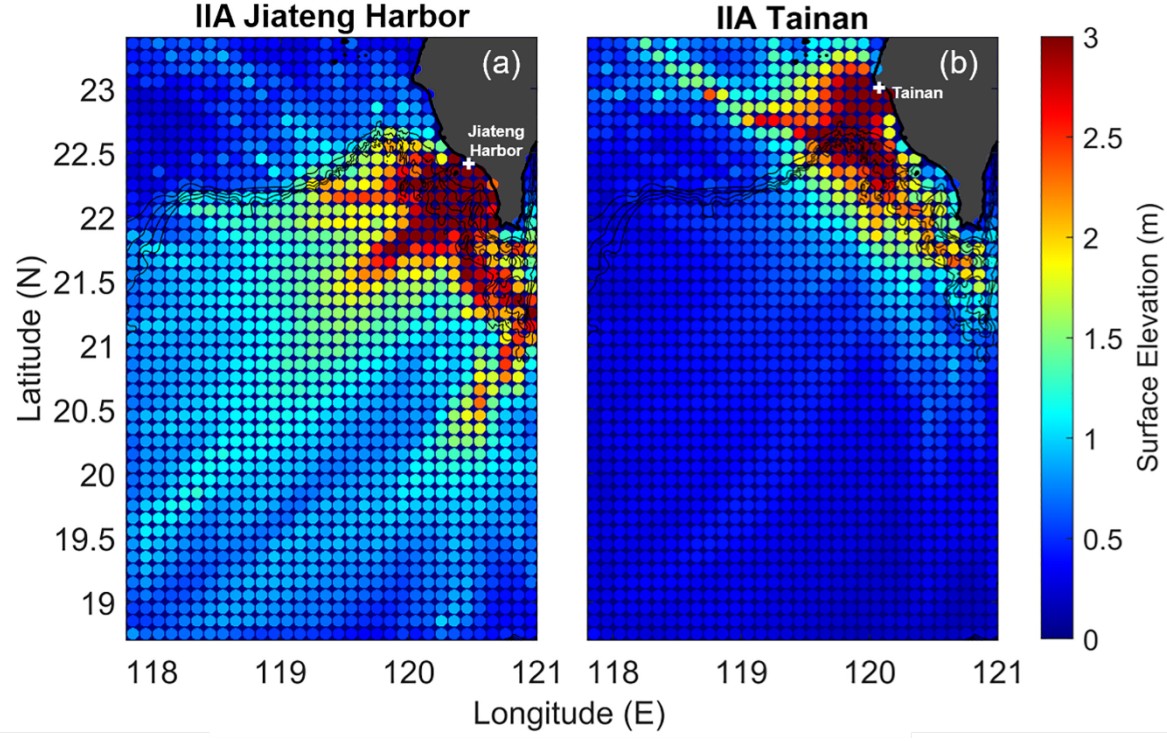

**Figure 3 IIA results of the southwestern coast of Taiwan. Panels (a, b) show the IIA result of Jiateng Harbor and Tainan as the white crosses denote the location of study sites, respectively. The circles represent the locations of unit-source tsunami, with colors showing the value of maximum wave height recorded at the study site.**

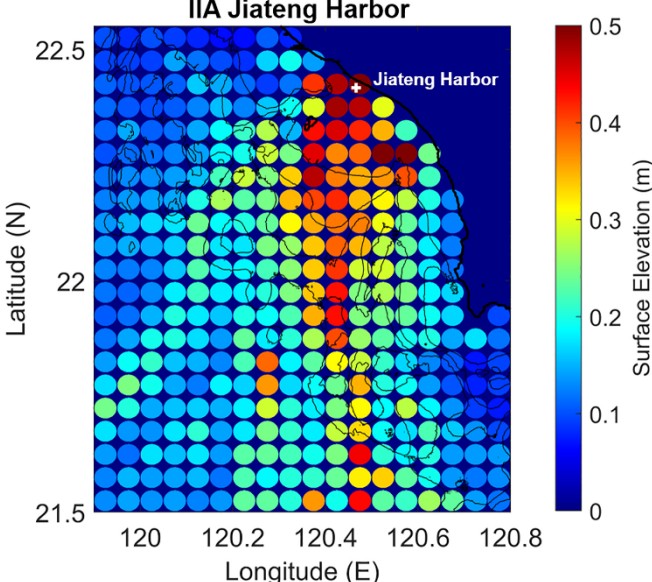

**Figure 4 High-resolution IIA results of the Jiateng Harbor. The circles represent the locations of unit-source tsunami. Color denotes the value of maximum wave height recorded at the study site.**

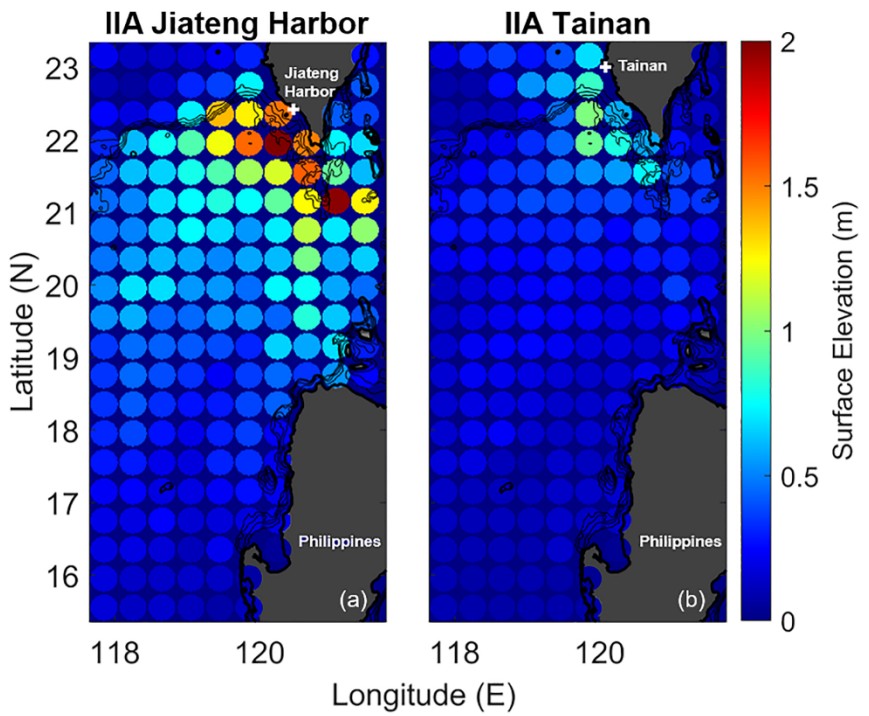

**Figure 5 IIA results of the far-field region.   Panels (a, b) show the IIA result of Jiateng Harbor and Tainan. The white crosses stand for the location of study sites, respectively.   The circles are the locations of unit-source tsunami, with colors showing the value of maximum wave height recorded at the study site.**

## 2.2 Modeling Approaches

In addition to the analysis method, tsunami scenarios were performed with COMCOT as well for the purpose of examining tsunamis effects on the coastal regions of Taiwan. Tsunamis caused by SMFs were designed referring to the results of IIA method and previous researches through the submarine morphology of Taiwan (Chen et al., 2018; Chiang and Yu, 2006; Chiang et al., 2020; Hsu et al., 2018; Liu et al., 1993; Su et al., 2018). Since having a reasonable initial free-surface elevation for the source is crucial, the amplitudes of SMF tsunamis (Table 2) were obtained by the length and thickness of the mass

failure along with the local water depth and slope (Watts et al., 2005)

$$\eta_{2D} \cong 0.0286T(1 - 0.750sin\theta)\left(\frac{b\ sin\theta}{d}\right)^{1.25} \tag{1}$$

where T and b denote the thickness and length of the mass failure, respectively, d represents the water depth at the center point of the mass failure, and $\theta$ is the slope angle.

**Table 2 Parameters of 1781 SMF Scenarios.**

| SMF Scenario | Fangliao Ridge (FR) | Fangliao Slide (FS) | Kaoping Canyon (KC) |
|---|---|---|---|
| Longitude (°E) | 120.56 | 120.40 | 120.23 |
| Latitude (°N) | 22.22 | 22.083 | 22.341 |
| Length (km) | 4.0 | 12.0 | 4.0 |
| Width (km) | 5.0 | 12.0 | 5.0 |
| Initial elevation (m) | 8.0 | 5.0 | 4.0 |

  In this research, besides the local SMFs tsunamis, seismic tsunamis generated by the deformation front, the northern part of Manila Trench, and the Hengchun Fault were chosen. Since the upper part of Penghu Canyon was highlighted by the results of IIA (Fig. 3), a scenario of $M_w$ 7.5 earthquake with focal mechanism of strike: 194°, dip: 20° and rake: 90° was designed

referring to the 1661 Tainan Earthquake (Han et al., 2017; Ma and Lee, 1997; Okal et al., 2011) for studying the impact of tsunamis generated at the deformation front. Additionally, tsunamis generated at the Manila Trench have been viewed as a threat to the South China Sea area (Li et al., 2016; Liu et al., 2009; Megawati et al., 2009; Okal et al., 2011; Qiu et al., 2019; Wu and Huang, 2009). Hence the upper part of the Manila Trench was selected to study the tsunami effect on the southwest part of Taiwan. Following Wu (2012) and Qiu et al. (2019), two scenarios of the Manila Trench were conducted to compare

the tsunamis generated with different seismic parameters.

  Wu (2012) presented a series of scenarios for estimating the tsunami hazard in Taiwan. The information of rupture area and slip were obtained with the source-scaling of Yen and Ma (2011). For the worst-case tsunami hazard assessment, Wu (2012) also applied the asperities, which are regions on the fault that have larger slips compared to the average slip on the rupture area (Somerville et al., 1999) in some of the tsunami sources in his study. In this research, we selected T02, the $M_w$ 8.2 Manila

Trench earthquake of Wu (2012) to study the tsunami effect.   On the other hand, two models (A and B) explaining the plate movements on the Manila megathrust and other faults on Luzon Island have been proposed by Hsu et al. (2016).   From the geodetic analysis, Hsu et al. (2016) found that the magnitude of earthquakes in the Manila subduction zone could reach $M_w=$ 9+ with an assumption of a 1000-year return period.   To assess the potential tsunami hazard of SCS, Qiu et al. (2019) selected scenarios which release 1000 years of accumulated strain and studied the plausible events.   Results showed that the model B

generated larger tsunami impacts than model A did in both near-source and far-field regions (Qiu et al., 2019).   Hence, the initial surface displacement of the Gaussian slip distribution with a 50 % coupling ratio scenario of Qiu et al. (2019) was chosen and modeled as well.   Meanwhile, although limited damage at the coastal area was expected when the inland earthquake was mainly generated at Hengchun Fault (Wu, 2012), the simulation was conducted based on the descriptions of the foreign reports of Jäger (1784) and Perrey (1862) ("subterranean movements causing the whole island to shake and be

devastated; the earthquake lasted for 8 hours") and its location at the southern tip of Taiwan (Lin et al., 2009).

    To model the evolution of tsunamis, nonlinear shallow water equations were used with a bottom friction of Manning coefficient valued 0.013 for SMFs and seismic scenarios.   Since SMFs tsunamis are generally local events, high-resolution bathymetry of 100 m was employed, focusing in the study area with 0.05 arcmin numerical resolution.   To model the evolution of tsunamis triggered by SMFs, nonlinear shallow water equations were used with a bottom friction of Manning coefficient valued 0.013.

On the other hand, nested-grids were set for the seismic tsunami scenarios.   The first layer was conducted with 1 arcmin bathymetric data of ETOPO1 from NOAA.   Covering Taiwan (119.10°E-122.50°E, 21.50°N-22.50°N), the second layer bathymetry was extracted from the data downloaded from Ocean Data Bank of the Ministry of Science and Technology (http://www.odb.ntu.edu.tw/) with 200 m resolution.   Finally, bathymetric data of the third layer was adopted from the fine coastal area data of Wu (2012).   Combining the topographic data from the Department of Land Administration of Ministry of

Interior and the Research Center for Space and Remote Sensing of National Central University, together with the bathymetry provided by Taiwan Ocean Research Center, the resolution of the third layer was interpolated to 27.5 m numerically from the original resolution of 0.0004 arc degree.

## 3 Results

### 3.1 The 1781 Jiateng Harbor Flooding

With the detailed descriptions documented in the report, it is possible to reassess this historical event.   From the sentences "Giant wave appeared and the water rose for tens of zhang high (1 zhang approximately equals to 3 - 1/3 m);  People were swinging on top of bamboos; In few quarters, the water retreated quickly;  The fisherman sailed on top of bamboos on raft; When the wave ebbed, the rafters of the thatched roofs were all gone", it was assumed the wave height that stuck Jiateng Harbor was about two meters, close to the wave height that could affect the coastal infrastructures (Yu, 1994) or even higher,

and ebbed in few quarters.

Li et al. (2015) considered that SMFs in the Kaoping slope region could have been the major contributor to the 1781/1782 tsunami event according to the available historical records at that time, the numerical results, together with the fact that offshore southwest Taiwan is a SMFs-prone environment. Submarine landslide tsunami scenarios were then designed to investigate and reconstruct the 1781 Jiateng Harbor Flooding.

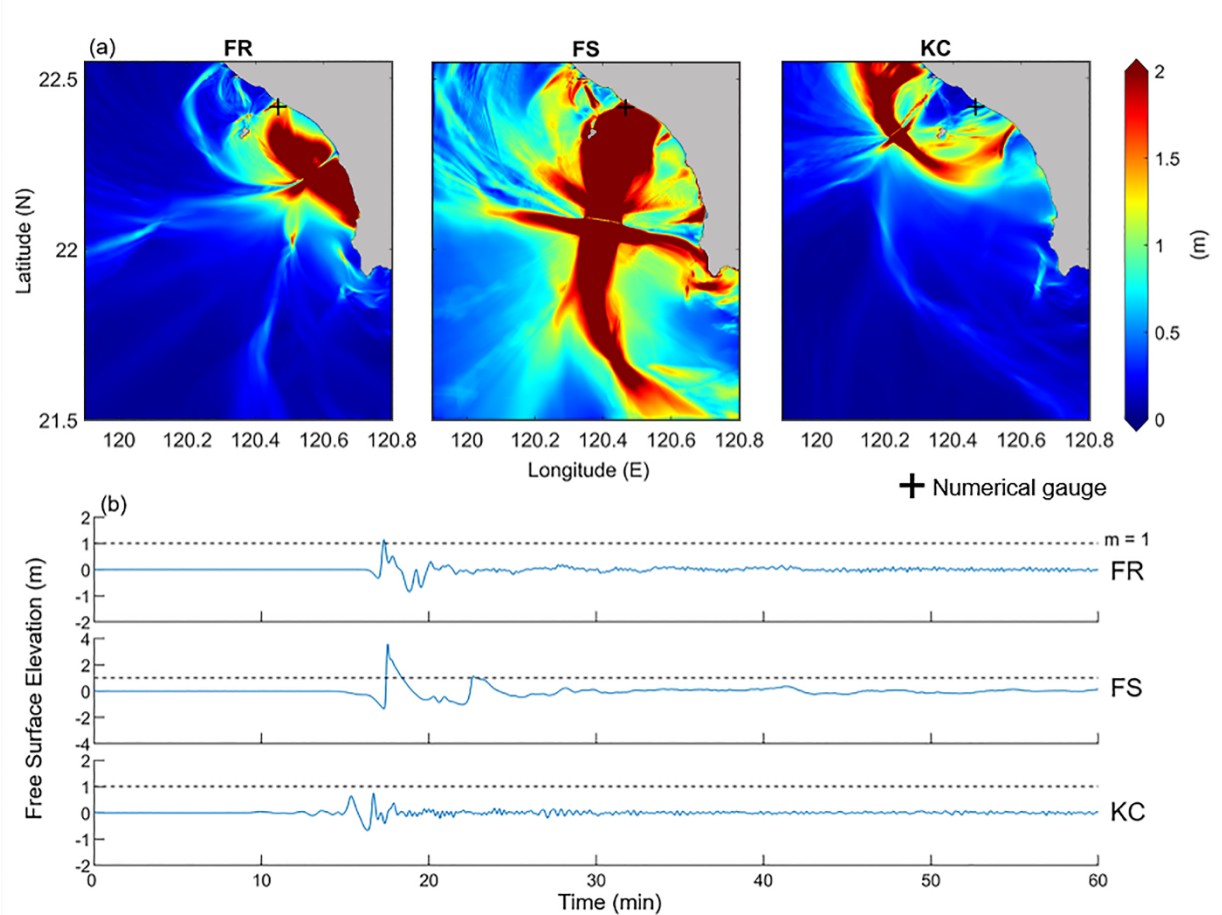

Figure 6 (a) Maximum wave height of the 1781 SMF tsunami simulations. The black cross denotes the numerical gauge of Jiateng Harbor (120.467°E, 22.417°N). (b) Numerical gauge records at the Jiateng Harbor study site for the SMFs scenarios of FR, FS and KC. One-meter elevation was marked with dashed lines for comparison.

The maximum wave heights of SMF tsunami scenarios (Fig. 6a) showed that the tsunamis with sources located at FR and FS traveled directly to the Jiateng Harbor study site, whereas the waves of KC traveled north of Kaohsiung. The numerical gauge (Fig. 1c) records of FR and FS held also higher values amid the scenarios (Fig. 6b). Moreover, the wave crest reached to 3.5 m for the simulation of FS. The results that the half cycles of first waves were all less than ten minutes, support the description "in few quarters" in the historical report of 1781 Jiateng Harbor Flooding.

**3.2 The 1782 Tsunami Event**

To investigate the 1782 Tsunami, we conducted four numerical scenarios in order to reproduce the devastating event that
match the historical reports described as "no part of the flooded island remains visible except for the foot of the mountains"
(Jäger, 1784; Perrey, 1862). During the simulation of the deformation front, which was generated by a $M_w$ 7.5 earthquake
near the Penghu Canyon, the tsunami wave mainly impacted on the coast of Tainan and Kaohsiung (Fig. 7a). However, the
wave heights recorded by the numerical gauges at the Jiateng Harbor and Tainan were less than 0.5 m (Fig. 8a, 8b), far from
matching the disastrous event described in the historical reports of the 1782 Tsunami event.

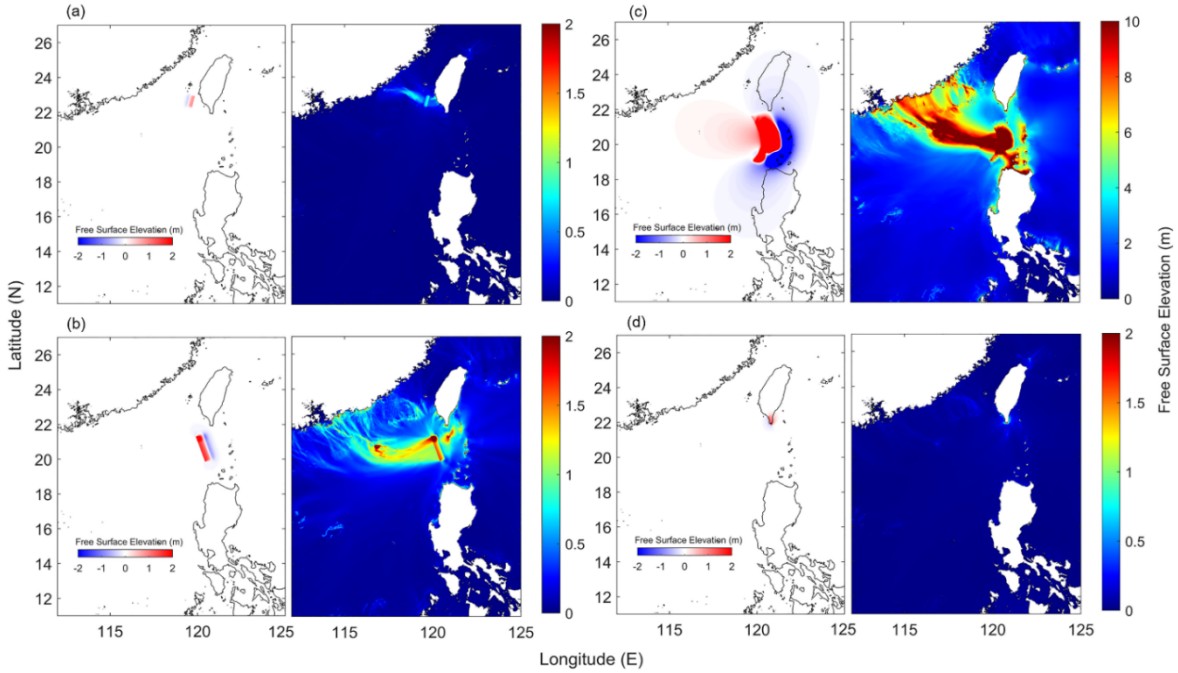

**Figure 7 The initial elevations and the maximum wave heights of the seismic tsunami scenarios of (a) the deformation front, (b and c) the Manila Trench and (d) the Henchun Fault.**

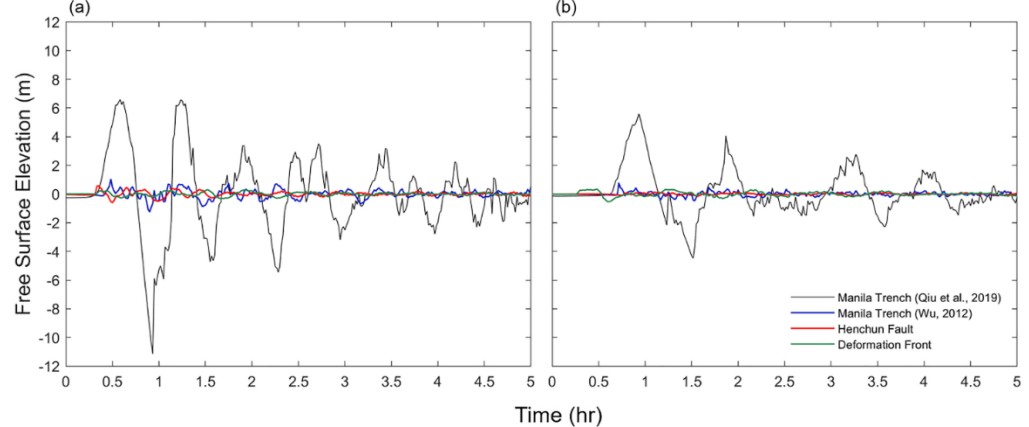

**Figure 8 The numerical gauge records at the study sites of (a) Jiateng Harbor and (b) Tainan.**

In contrast with the scenario of the Deformation Front, the results of the Manila Trench indicated how destructive tsunami scenarios could be. In particular, massively impacted areas include the northern Philippines, southeast China and southern part of Taiwan (Fig. 7b, 7c). In Fig. 8, the surface fluctuations reached the height of 1.02 m at the Jiateng Harbor and of 0.76 m at Tainan with the parameters of Wu (2012), yet the study sites were impacted by tsunami waves over five meters in the other scenario of the Manila Trench. With the initial surface elevation of Qiu et al. (2019), maximum wave height of the tsunami appeared over five meters along the coastline of Tainan, Kaohsiung and Pintung (Fig. 7c). Time series of the numerical gauges showed that tsunami waves arrived at the Jiateng Harbor study site after 20 minutes, impacting with a wave height of 6.59 m at first, then the second wave arrived after 1 hour and 10 minutes with a height of 6.56 m (Fig. 8a). On the other hand, the first wave arrived at Tainan 39 minutes after the tsunami occurrence. The wave height recorded was 5.6 m and the height of the second wave was reduced to 4.06 m (Fig. 8b). The fluctuations of the free surface decayed to less than one meter after 4.5 hours, about half of the time (8 hr) described in the historical reports.

Finally, fault parameters of the Henchun Fault (Wu, 2012) were employed to investigate the 1782 with sensible seismic movements (Fig. 7d). Limited tsunami waves generated by the $M_w$ 7.6 Henchun Fault earthquake was expected for its location. Although parts of the Pingtung coast faced tsunami waves over three meters high, the maximum wave height was less than one meter in Tainan and less than two meters in Donggang area. Tsunami waves recorded by numerical gauges at both study sites of the Jiateng Harbor and Tainan didn't exceed 0.6 m (Fig. 8a).

## 4 Discussion

The findings from historical reports suggest that the 1781 Jiateng Harbor Flooding and the 1782 Tsunami should be two independent incidents. Based on a comprehensive investigation of the historical reports, it is conspicuous that the descriptions are pointing to different events although the time of year is close. First of all, there is not any seismic movement mentioned in the Chinese record of 1781 Jaiteng Harbor Flooding (Chen, 1830) while shaking movements and tremors are documented in the German report for the 1782 incident (Jäger, 1784). Secondly, there are different descriptions on the severity of the resulted damage. It is recorded that the whole island of Formosa was inundated by ocean water till the foot of the mountains and forty thousand people were killed in the 1782 Tsunami event (Jäger, 1784). However, despite the fact that the coastal area was impacted by the waves, only one person died in the 1781 Jiateng Harbor Flooding (Chen, 1830).

The results of numerical simulations performed with COMCOT seem to indicate that the 1781 Jiateng Harbor Flooding and 1782 Tsunami are two different events. In the SMFs study, tsunamis generated in southwest Taiwan, especially from the South-South West direction, could be the main source of the 1781 Jiateng Harbor Flooding. The wave height of the FS scenario fits best the historical description, in particular the half cycle less than 10 minutes and the duration of "in few quarters". Nevertheless, the SMFs alone could hardly generate tsunamis that submerge Taiwan. In this study, the disastrous scenario of Qiu et al. (2019) suggests that tsunamis triggered at the Manila Trench could contribute to the 1782 Tsunami event. However, historical records are not found for the tsunami waves in the southeast China and northern Philippine in a time

compatible with the event.  Moreover, it is unlikely that a severe tsunami in 1782 was not reported by any Chinese document. Therefore, the existence of the 1782 Tsunami is less possible and remains doubtful.

Li et al. (2015) were the first group who read, translated and reported the French record as documented by Perrey (1862).  In this study, for the first time the historical documents (Gazette de France, 1783; Jäger, 1784) cited by Perrey (1862) are fully translated and reported together.  By analyzing the historical records, the 1782 Tsunami event is clarified not to be a

typographical error of the 1867 Keelung Tsunami proposed by Lin (2006) since the literature records (Gazette de France, 1783; Jäger, 1784; Mallet, 1854; Perrey, 1862) had existed already before the year of 1867.  Incoherence was found amid historical reports of the 1782 Tsunami event since seismic movements were not mentioned in the text of Gazette de France (1783) and Mallet (1854).  While examining the historical documents, records of Griffiths (1785), Grosier (1785) and Davidson (1903), which reported the storm event on the 22 May of 1782, also put the existence of the 1782 Earthquake in doubt by following

the procedure of validation of Mak and Chan (2007).  Together with Chinese records and foreign documents of the year 1782 (Davidson, 1903; Griffiths, 1785; Hsu, 2007; Rees, 1820; Yin, 2017), the fact that the significant water level was determined by the storm surge should not be ignored.  Meanwhile, since active submarine mud volcanoes were observed in the offshore area of southwest Taiwan (Chen et al., 2014; Mai et al., 2021), we shall neither rule out the possibility of multiple hazards occurring coincidently or very close in the same period (e.g. The Eruption of Mount Pinatubo and Typhoon Yunya in 1991).

Therefore, numerical simulations of storm surges or complex hazards should be taken into consideration in future studies.

This research suffers some limitations, including the lack of gauge records of the two incidents and the evolution of coastal bathymetry and topography.  In the past 200 years, the coastline of southwest Taiwan has changed a lot due to the sediment transportation and the erosion (Chang et al., 2018).  Without knowing the exact bathymetry in the 18th century, the accuracy of the numerical gauges may not be satisfied.  Therefore, the figures of the maximum wave height, with which the tsunami

flux propagation could be traced, should be taken into consideration in the first place.  Moreover, the landslide tsunamis require SMF dating to confirm the sources.  Hence, further marine surveys are indispensable for fully reconstructing the 1781 Jiateng Harbor Flooding.

Examining the historical events allows us to assess the tsunami hazard and to prepare for natural disasters that could possibly occur in the future.  Results of the study based on the IIA method show that the offshore region of southwest Taiwan and the

Luzon Trench could generate tsunamis that affect most to the southwestern coast of Taiwan.  In fact, tsunamis generated locally could impact the coastal area within a very limited time.  Hence, a proper evacuation plan is needed for the residents at the seaside.

**Appendix A: Translations of Gazette de France (1783) and Jäger (1784)**

**Text A1**

Original text from Gazette de France.: De Paris, le 12 Août 1783.

Une lettre de la Chine fait mention d'un évènement arrivé l'année dernière, & peut-être plus terrible encore que ceux qu'ont éprouvés la Sicile & la Calabre dans le commencement de celle-ci féconde en désastres. En attendant une relation plus détaillée, voici ce que l'on en raconte : Le 22 Mai de l'année dernière, la mer s'éleva sur les côtes de Fo-Kien à une hauteur prodigieuse, & couvrit presqu'entièrement pendant huit heures l'île de Formose qui en est à 30 lieues. Les eaux, en se retirant, n'ont laissé

à la place de la plupart des habitations que des amas de décombres sous lesquels une partie de la population immense de cette Isle est restée ensevelie. L'Empereur de la Chine, voulant juger par lui-même des effets de ce désastre, est sorti de sa capitale ; en parcourant ses provinces, les cris de son Peuple excités par vexations de quelques Mandarins, ont frappé ses oreilles ; & on dit qu'il en a fait justice en faisant couper plus de 300 têtes.

(Translated from French to English by T.-C. Liu, June 2020)

A letter from China mentions an event that happened last year, and perhaps it's more terrible than those experienced in Sicily and Calabria, where suffers disasters a lot in the beginning. Pending a more detailed narrative, here's what they tell us: On May 22, 1782, the sea level rose to a prodigious height on the coast of Fujian, covering almost the entire island of Formosa for 8 hours, which situated 30 league (1) away. The water ebbed, leaving most of the residential areas nothing but piles of rubble, under which a part of the huge population stayed buried. Wanting to inspect the effects of this disaster by himself, The Emperor

of China left the capital. During the journey going through his provinces, the cries of his People, whipped up by the irritation of some Mandarins, struck his ears, and it is said that he executed over 300 people.

(1) The ancient measurement unit "lieue" is about 4 km long in length.

**Text A2**

(Translated from German to English by D. David, June 2020)

§116

Following the changed order [of the text] mentioned in Scholion [i.e. commentary] of §112, the submerging of the Island of Formosa into the sea (1) will constitute the third point.

The French Foreign Minister [Ministre d'État] Bertin (2) received a letter from a missionary in Beijing, which contained the most unfortunate news that, in October of 1782, on the Island of Formosa, situated right off the coast of China, fire-breathing

mountains and maws unexpectedly burst open. Accompanying their harrowing eruption were subterranean movements, causing the whole island to shake and be devastated. The ocean waves, which were pushed from east to west, evenly covered the whole of the submerged island so that no part of the flooded island remains visible except for the foot of the mountains. The shaking movements and tremors continued for more than eight hours. The three most noble cities of this island (whose names we cannot inquire about on the submerged island) alongside twenty market towns are buried beneath the rubble and

their remains were carried away by the force of the sea. More than forty thousand residents, some of them native, some of them Chinese, who found their graves in this accident, have been counted. All promontories that had projected into the sea have been washed away and the locations where they had been situated have now been turned into reservoirs and dregs of water. The forts Seeland (3) and Pingschingi as well as the hills they were built on have vanished. In short, everything has been leveled by the water. Formosa in 1782 and Messina in 1783 (4) have shared the same destiny and both have seized to be

what they once were.

*Scholion.* The Island of Formosa is located in the Asiatic Sea, between Japan, China, and the Islands of the Philippines, twenty-four miles off the Chinese coast. It is about seventeen miles in length and fifteen miles in width. Just like Piccando and Legneto it is very fertile and rich in gold and silver ore. In the past, this island had its own king, up until around two hundred years ago when it was conquered by the Tartars (5). And after it had rid itself of their yoke, it was subdued by the Japanese sixty years

ago. Of the Europeans, especially the Dutch have settled here and on the small isle of Tyowan built the Fort Seeland, which was in their possession for a long time until it was taken from them in 1661 by the Chinese pirate Coxinga (6).

§117

The information designating the time when this disaster struck the island is not consistent. Then, a letter comes to mind supposedly sent to the former Minister of War Monsieur Bertin from Beijing by a Chinese who had been to Paris many years

ago. Therein it states: "In December of 1782, various fire-breathing mountains on the Island of Formosa began to open their fiery jaws. Their dreadful eruption was accompanied by a massive subterranean movement, which shook the whole island and let it be submerged by the seas rolling from east to west. The tremor lasted longer than eight hours. More than forty thousand residents have been counted that found their graves in this horrific accident."

And yet another message reported the following: A second letter from Beijing arrived in Versailles that confirmed everything

that had already been reported on the disaster of the Island of Formosa. The suffering of a few thousand who had escaped the stormy waves is said to be unspeakable—as one can imagine.

The emperor of China is said to have sent the following letter to the viceroy of the region of Feu-kim (7) which we indent here because of its rather strange content (one may doubt whether this letter was really written by the emperor): "I have heard of the disaster that has befallen my Island of Ray-Onan (i.e. Formosa) on May 22, 1782. I hence order you to inquire in detail

about the damages the remaining inhabitants of this unfortunate island were forced to suffer and report back to me immediately so I can swiftly render aid. The homes and buildings devastated by the water shall be rebuilt at my expense, and those damaged shall be repaired. You shall provide those unfortunate people with all necessary food at my cost. You shall afford everyone with my help without exception. It would pain me if you neglected just one of them. They should know that I am watching over all of them and that I love all of them fondly. You shall tell them that it is me who is helping them, their lord and father.

Also, you shall use funds from treasury to rebuilt as many warships and magazines as this almighty arm has taken from me through the storm and the waves. Do not shirk [from your duty], I forbid it, and report back to me how you have obeyed my will."

*Scholion.* Because three different months have been given, namely the months of October, December, and May, the actual timing of when the Island of Formosa was destroyed cannot be determined. It is enough for us to know the year. And from the letter sent by the missionary from Beijing to Monsieur Bertin in the year of 1783 we can gather that during the eruption of the subterranean fires, the sea on the coast of China towered to an unnatural or unusual height and flooded and submerged the whole island so that not the tiniest bit could be seen for several days. And after the water had retreated, one could make out no sign at all, not of the human population nor of any four-footed animals.

In all of history, one cannot find an incident so horrific like the ones that happened in Formosa and Messina.

(1) The German "Untergang" may also refer to "doom," "demise," "downfall." In this context, a sinking or submerging into the sea seems more appropriate.

(2) Henri Léonard Jean Baptiste Bertin (1720-1792).

(3) Fort Zeelandia, built in 1634 by the Dutch East India Company (VOC) in Anping.

(4) Reference to the 1783 Calabrian earthquakes.

(5) Reference to the Manchus, i.e. the Qing Dynasty.

(6) Zheng Chenggong (1624-1662).

(7) Probably a variant spelling of Fukien or Fujian Province. During the Qing Dynasty, the Governor-General of Taiwan, Fujian and Zhejiang was known as the Viceroy of Min-Zhe.

**Author contribution**

TRW conceptually designed the study and supervised the research; TCL performed and analyzed the numerical simulations as well as wrote the paper; SKH provided the fine bathymetry data and provided constructive suggestions to the study.

**Competing interests**

The authors declare that they have no conflict of interest.

**Acknowledgements**

We would like to gratefully thank Dr. Qiang Qiu and Dr. Linlin Li, for kindly providing the initial surface elevation data of the numerical simulation and for their helpful comments about this research. We also acknowledge the Editor, Dr. Alberto Armigliato, and the two anonymous reviewers for the valuable and constructive suggestions that have significantly improved the quality of this paper.

**Financial support**

This study has been supported by the Taiwan Ministry of Science and Technology (MOST; grant no. MOST 108-2116-M-008-017 and MOST 110-2111-M-008-017), the Seismology Center of Central Weather Bureau (grant no. MOTC-CWB-110-E-04), as well as the Center for Astronautical Physics and Engineering (CAPE) and Earthquake-Disaster & Risk Evaluation and Management Center (E-DREaM) from the Featured Area Research Center program within the framework of Higher Education Sprout Project by the Taiwan Ministry of Education (MOE).

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
