# Peer review of "Historical Tsunamis of Taiwan in the Eighteenth Century: the 1781 Jiateng Harbor Flooding and 1782 Tsunami Event"

_Natural Hazards and Earth System Sciences, 2021_

## Author Comment (AC1)

Dear reviewer,

We sincerely acknowledge the reviewer for the careful reading, helpful comments, and constructive advises. The manuscript would be revised according to all the suggestions in order to improve the quality. Responses to each comment could be found in the following part.

**Major comment:**

In my opinion, a brief but exhaustive description of the IIA method in section 2 is required. Indeed, there is no description about the way in which the maximum elevations are calculated starting from the unit sources. All the references to previous works based on this method (Chung, 2018; Lee, 2014; Wu, 2017), as reported in the text, are in Chinese language only. If this method is unpublished in international peer-reviewed journals, it should be exhaustively described here.

Reply: We acknowledge the reviewer for reminding the necessity of introducing the IIA method comprehensively for publishing in the international peer-reviewed journal. Follower the reviewer's suggestion, the section 2.1 would then be enlarged with descriptions of the IIA method in greater detail in the new revision.

**Moderate comments:**

1) I recommend to clarify already in the abstract that these two events are tricky not only for the difficulties in interpreting the existing documentation of both, but also because they resulted close enough in time and location to have raised the suspect, in some researchers, to be the same event.

This, at least to me, would make clearer also why the authors presented the two events together.

A short sentence before "Reasoning these historical events [...] " in the abstract could serve the scope.

Reply: We thank the reviewer for the recommendation.    Pointing out in the abstract that these two events were close in time and location would certainly help explain why we present them together in this study.    As the reviewer suggested, the sentence "These two events seem to be close enough in time and location that, to some researchers, they are considered as the same event" would be added before "Reasoning these historical events [...]" in the abstract.

2) I recommend to add a Figure (a new Figure 1) to introduce the area of study, with a zoom in the two targets, in order to make the reader aware, since the beginning, of the regional context and the distances between the two sites. In this new figure, I suggest also to include the location of some features described later on in the text (canyons, faults, etc) in order to provide a reference between the real geography of the area and what described in the text.

Reply: We agree with the reviewer and apologize for not being thoughtful enough to the readers.    A new Figure 1, which includes the submarine features off southwest Taiwan and the zoom-ins of the two study areas, would be added.

3) The references for the bathymetry data with resolution 100 m used for the two near-field models should be provided (line 96). I suppose (but I am not sure) that this is different from what described for the data used with the nested-grids in COMCOT (lines 173-179).

Reply: We thank the reviewer for the reminder.    The bathymetry data with resolution 100 m (line 96 and line 169) provided by Prof. Shu-Kun Hsu, one of the co-authors, are gained fundamentally through compilations of data from different marine cruises.    They different from the bathymetric data used for the nested-grids in COMCOT (lines 173-179).

Hsu, S.-K., Sibuet, J.-C., Monti, S., Shyu, C.-T. and Liu, C.-S.: Transition between the Okinawa trough backarc extension and the Taiwan collision: new insights on the southernmost Ryukyu subduction zone, Mar. Geophys. Res., 18(2), 163-187, doi: https://doi.org/10.1007/BF00286076, 1996.

Hsu, S.-K., Liu, C.-S., Shyu, C.-T., Liu, S.-Y., Sibuet, J.-C., Lallemand, S., Wang, C. and Reed, D.: New gravity and magnetic anomaly maps in the Taiwan-Luzon region and their preliminary interpretation, Terr. Atmos. Ocean. Sci., 9(3), 509-532, doi: 10.3319/TAO.1998.9.3.509(TAICRUST), 1998.

4) I did not understand why the high-resolution IIA (Figure 3) is shown for the Jiateng target only. What about Tainan?

Reply: According to the descriptions of historical reports and the previous study of Li et al. (2015), a local submarine mass failure could also be responsible for the Jiateng Harbor Flooding.   We then performed 378 unit-source tsunamis scenarios, where high-resolution bathymetry is required in order to better describe the regional submarine features, to form the near-field IIA.   On the other hand, the descriptions of 1782 Tsunami Event is different in the severity that the ocean waves "covered the whole of the submerged island so that no part of the flooded island remains visible except for the foot of the mountains" (Jäger, 1784).   Referring to the historical report above, it seemed that the 1782 Tsunami Event was affecting the whole island of Formosa instead of being limited to the local area of Tainan.   Therefore, we applied IIA method in larger areas (the southwestern coast and the far-field one).

5) What the sentence "the asperity effect was also applied" (line 155) means in practice? At least a brief sentence about this should be provided.

Reply: We thank the reviewer sincerely for the comment.   According to Somerville et al. (1999), "An asperity is a region on the fault rupture surface that has large slip relative to the average slip on the fault".   Wu (2012) mentioned that "the asperity is not directly connected to the tsunami earthquake. However, the tsunami height will be enlarged, if the asperity is located in an offshore area or on the tsunami ray".   In order to assess the worst-case tsunami hazard of Taiwan, Wu (2012) applied then the asperity effect to some of the tsunami sources in his study, including T02, the one we selected and used in this study.

The sentence of line 156 would be rephrased as below:

For the worst-case tsunami hazard assessment, Wu (2012) also applied the asperities, which are regions on the fault that have larger slips compared to the average slip on the rupture area (Somerville et al., 1999) in some of the tsunami sources in his study.

With the following reference of Somerville et al. (1999) added.

Somerville, P., Irikura, K., Graves, R., Sawada, S., Wald, D., Abrahamson, N., Iwasaki, Y., Kagawa, T., Smith, N. and Kowada, A.: Characterizing crustal earthquake slip models for the prediction of strong ground motion. Seismol. Res. Lett., 70(1), 59-80, doi: 10.1785/gssrl.70.1.59, 1999.

6) At line 170, it seems that nonlinear equation were used for SMF only, but I guess that they have been used for the seismic sources as well. Authors should better clarify this point. Also, what is the reason for the selected Manning coefficient value (line 171)? 0.013 is a quite low friction, often used in presence of plain surfaces and no obstacles.

Reply: We thank the reviewer for the comment and apologize sincerely for missing this key information for the seismic tsunami sources. The nonlinear shallow water equations were indeed used for the seismic tsunami scenarios as well. We would add the clarification and more information in this paragraph (line 169-179).

Indeed, just like the reviewer points out, 0.013 is a Manning coefficient value relatively low. We are using this value as a conservative estimate since we don't really know what happened in the historical events, yet it seems to us a bit reckless not to apply any friction or higher friction values for the tsunami simulations. Also, most of the southwest coast of Taiwan are fine-sand beaches. Finally, the locations of Jiateng Harbor and Tainan are close to the lagoons which exist already in the Qing Dynasty. According to the reasons above, we conservatively selected the Manning coefficient value 0.013 for the numerical simulations of this part.

7) At line 183 is reported: "1 zhang approximately equals to 3 - 1/3 m" but I am not sure to have understood which is the exact correspondence in meters.

Reply: We thank the reviewer for raising this question. Actually, the length unit mentioned here confused us as well, though the original historical report is written in

Mandarin (Li et al., 2015).    Therefore, we suggest here using the villagers' houses and the bamboos as the height reference for estimating the tsunami height impacting the Jiateing Harbor.

Since "the rafters of the thatched roofs were all gone" (Li et al., 2015), and most of the residents at that time live in one-floor houses, the tsunami height was estimate to be higher than the houses of the villagers.    Then according to Yu (1994) and the English translation of the supplementary element of Li et al. (2015), people were climbing upward (in the text, we know people climbed up the bamboos) as the water level rises.    Supposing that the reaction time and the climbing-up time was probably very limited for the residents, and after the flooding there was "One strong man jumped to ground, and helped others getting down" (Li et al., 2015), with the height that villagers could climb up within the limited reaction time and the height that one can jump down safely, the possible tsunami height estimated from the description is about two meters or higher in this study.

8) Section 3.1: I suggest to provide some more details on the position of the numerical gauge, as the water depth of the point and a zoomed map with the position respect with the coastline (an inset in Figure 5 could be enough)

 Reply: We acknowledge the reviewer for the suggestion.    Considering that the gauge locations are mentioned already in the section 2.1, we would add two zoom-in maps with the water depth of the gauge points to the "current" Figure 1 since it is equally important to show more information of both the numerical gauges of the Jiateng Harbor and Tainan.

9) At line 235 I suggest to change the sentence in "[...] COMCOT seems to confirm [...]".

I would be more prudent on what the modelling results are indicating, since the parameters of the potential sources are quite uncertain for these events of the past. Modelling a few scenarios can help to support some hypotheses in broad terms, but I would be careful in drawing definitive conclusions.

 Reply: We thank the reviewer for the suggestion.    We should be more careful while making the conclusion.    The sentence of line 235 would be rephrased as "The results

of numerical simulations performed with COMCOT seem to indicate that the 1781 Jiateng Harbor Flooding and the 1782 Tsunami are two different events".

**Minor suggestions:**

line 12: "[...] , titled Taiwan Interview Catalague, [...]"

 Reply: We thank the reviewer for the comment. The sentence in the abstract would be rephrased as "The 1781 Jiateng Harbor Flooding, recorded by the Chinese historical document, titled Taiwan Interview Catalogue, took place on the southwest coast of Taiwan".

line 70-74: I suggest to slightly rephrase these sentences, since it seems to me that the reader could be a bit confused. I suggest something similar to the following: "Second, from the words "Voici ce que je lis dans J. L. Ab Indagine L. M." (i.e., "Here is what I read from J. L. Ab Indagine L. M.") in Perrey (1862), it is reasonable to believe that most of the content in that document is quoted from the German report Philosophisch und physikalische Abhandlungen (Jäger, 1784). After examining these two documents, it is also suspected that the date of a second letter sent from Beijing to Versailles was reported with a typographical error about the year. Indeed, the date "En décembre 1682" is found in Perrey's [...] "

 Reply: We agree with the reviewer and sincerely acknowledge the reviewer for the helpful suggestion. The sentences would be rephrased accordingly.

line 106: "[...] where the submarine structures are located, [...]" ?

 Reply: We thank the reviewer for the comment. The sentence in line 106 would be rephrased as "The areas where the submarine structures are located, [...]" as the reviewer suggested.

line 120: "broader"?

 Reply: We apologize for the typo. The sentence would be corrected accordingly.

line 167: I suggest to use quotation marks for the sentence ("subterranean movements causing the whole island to shake and be devastated; the earthquake lasted for 8 hours"), in order to emphasize the quotation.

 Reply: We acknowledge the reviewer for the suggestion. The quotation marks would be added to the sentence for emphasizing the quotation.

line 242: If I understand the sentence, I would rephrase as "Moreover, it is unlikely that a severe tsunami in 1782 was not reported by any Chinese document."

 Reply: We thank the reviewer for rephrasing the sentence, which makes the sentence much more understandable.    The sentence would be rephrased accordingly.

line 244: "[...] the French record as reported by Perry (1862) [...]"

 Reply: As the reviewer suggested, the sentence of line 244 would be rephrased as "Li et al. (2015) were the first group who read, translated and reported the French record as reported by Perrey (1862)".

line 245: "[...] the historical documents cited by Perry [...]"

 Reply: Following the reviewer's suggestion, the sentence would be rephrased as "In this study, it is the first time that the historical documents (Gazette de France, 1783; Jäger, 1784) cited by Perrey (1862) are fully translated and reported together".

---

## Author Comment (AC2)

Dear reviewer,

We sincerely acknowledge the reviewer for the careful reading, helpful comments, and constructive advises.    The manuscript would be revised according to all the suggestions in order to improve the quality.    Responses to each comment could be found in the following part.

Major comments:

1.  The introduction section can be better structured. Instead of listing the historical records and previous studies, some scientific questions/current issues could be raised in the very beginning. In the end of the introduction, readers may expect a brief explanation of the content in the following sections.

Reply: We acknowledge the reviewer for the constructive suggestion.    We would restructure the introduction and add a brief explanation of each section's content in the revised manuscript.

2.  The authors mentioned the possibility of storm surges could not be ruled out, one paragraph reviewing the most serious impact posed by past storm surge events could be very helpful for readers to understand the potential magnitudes.

Reply: We thank the reviewer sincerely for the suggestion.    Helping the readers to have a picture of how catastrophic that storm surge events could be is important. Following the reviewer's suggestion, the paragraph reviewing the storm surge cases would be added in the introduction for the new revision.

3.  After reading the newly added historical records in German, the possibility of volcanic eruption cause should be more carefully reevaluated, especially after the Tonga volcanic tsunami which occurred in Jan 15, 2022.

Reply: We agree with the reviewer. We would consider adding the volcanic part in the introduction and address it in the discussion part more carefully in the new revision.

Minor comments:

1. Line 32, a more detailed historical records should be added in the reference, Lau et al., 2010, NHESS, Written records of historical tsunamis in the northeastern South China Sea – challenges associated with developing a new integrated database.

   Reply: We acknowledge the reviewer for sharing the relevant literature. We would include the research of Lau et al. (2010) for historical tsunami records with greater details.

2. Line 40 in the Introduction, "Qin dynasty" should be "Qing dynasty"

   Reply: We apologize for the typo. The sentence in the line 40 would be corrected accordingly.

3. Line 120, "boarder"-"broader"

   Reply: We apologize for the typo. The sentence in the line 120 would be corrected accordingly.

---

## Author Response (AR1)

Responses to the two anonymous reviewers are presented below.
Reply to the Reviewer No.1: p1-p18
Reply to the Reviewer No.2: p19-p27

Reply to the Comments (RC1) of Reviewer No.1

Dear reviewer,

We sincerely acknowledge the reviewer for the careful reading, helpful comments, and constructive advice. The manuscript has been revised according to the reviewer's suggestions in order to improve the quality. Point-by-point reply to each comment could be found in the following part. The reviewer's comments are addressed in blueprints while we use the black and red prints to respond and mark the parts modified. The number of lines for the changing part is referred to the second version of the manuscript. All relevant changes are listed again at the end of this response file in line order.

**Major comment:**

In my opinion, a brief but exhaustive description of the IIA method in section 2 is required. Indeed, there is no description about the way in which the maximum elevations are calculated starting from the unit sources. All the references to previous works based on this method (Chung, 2018; Lee, 2014; Wu, 2017), as reported in the text, are in Chinese language only. If this method is unpublished in international peer-reviewed journals, it should be exhaustively described here.

Reply: We acknowledge the reviewer for reminding the necessity of introducing the IIA method more comprehensively for publishing in the international peer-reviewed journal. Follower the reviewer's suggestion, the section 2.1 has been enlarged with descriptions of the IIA method in better detail.

Changes made (line 116):

Unit-source tsunamis have been used for tsunami early warning and forecasting studies (Greenslade and Titov, 2008; Horspool et al., 2014; Liu et al., 2009; Matias et al., 2013; Percival et al., 2011). Since the IIA method aims to compare the effect of same-size

tsunamis generated at different locations, the shape of unit sources was therefore set up as a cylinder to ensure the isotropy of initial elevation. With the unit sources distributed evenly, the analysis method enables us to examine thoroughly the study areas. The IIA method is performed with the Cornell Multigrid Coupled Tsunami Model (COMCOT) (Liu et al., 1998; Wang and Power, 2011) and the process is presented with an example in Fig.2. As unit-source tsunamis simulations being conducted individually in the discretized region (Fig. 2a), we preserved only the initial elevation and the maximum wave height (Fig.2b, 2c) for computational efficiency. The maximum wave height at the assigned location would be extracted from each numerical scenario and colored at its source location, forming a map which levels the impact of tsunamis to the study site in the specific area (Fig. 2d).

[Figure]

**Figure 2 IIA method procedure. (a) Unit-source tsunamis are set in the discretized domain of the numerical simulation and conducted individually. (b, c) The results of the first unit-source tsunami are presented here as an example. The initial elevation and the maximum wave height are the essential data that should be saved for each simulation. (d) Each simulation's source location is colored by the maximum wave height recorded at the study site (marked by the white cross). Together they form a map leveling the impact of tsunamis in the study area.**

**Moderate comments:**

1) I recommend to clarify already in the abstract that these two events are tricky not only for the difficulties in interpreting the existing documentation of both, but also because they resulted close enough in time and location to have raised the suspect, in some researchers, to be the same event.

This, at least to me, would make clearer also why the authors presented the two events together.

A short sentence before "Reasoning these historical events [...] " in the abstract could serve the scope.

Reply: We thank the reviewer for the recommendation. Pointing out in the abstract that these two events were close in time and location would certainly help explain why we present them together in this study. As the reviewer suggested, the sentence "These two events seem to be close enough in time and location that, to some researchers, they are considered as the same event" would be added before "Reasoning these historical events [...]" in the abstract.

Changes made (line 15):

[…] These two events seem to be close enough in time and location that, to some researchers, they are considered as the same event. Reasoning these historical events requires carefully examining the literature records and performing the scenarios that match the descriptions. […]

2) I recommend to add a Figure (a new Figure 1) to introduce the area of study, with a zoom in the two targets, in order to make the reader aware, since the beginning, of the regional context and the distances between the two sites. In this new figure, I suggest also to include the location of some features described later on in the text (canyons, faults, etc) in order to provide a reference between the real geography of the area and what described in the text.

Reply: We agree with the reviewer and apologize for not being thoughtful enough to the readers. A new Figure 1, which includes the submarine features off southwest Taiwan and the zoom-ins of the two study areas, has been added.

Changes made (Page 2):

[Figure]

**Figure 1 The area of study.   (a) Bathymetry and related structures off South Taiwan with 250 m contour interval. (b, c) Zoomed maps of Tainan and Pingtung (i.e. Jiateng Harbor location). The contours are plotted with 20 m and 50 m, respectively. Red triangles denote the numerical gauges in this study and the gauge information are marked in the right corner of each figure.**

3) The references for the bathymetry data with resolution 100 m used for the two near-field models should be provided (line 96). I suppose (but I am not sure) that this is different from what described for the data used with the nested-grids in COMCOT (lines 173-179).

Reply: We thank the reviewer for the reminder.    The bathymetry data with resolution 100 m (line 96 and line 169) provided by Prof. Shu-Kun Hsu, one of the co-authors, are gained fundamentally through compilations of data from different marine cruises.    They are different from the bathymetric data used for the nested-grids in COMCOT (lines 173-179).

Changes made (line 130):

To ensure the accuracy of bathymetry settings, the bathymetric data with a resolution of 100 m which were gained fundamentally through compilations of data from different marine cruises (Hsu et al., 1996; Hsu et al., 1998), were adopted for the first two models.

References added:

Hsu, S.-K., Sibuet, J.-C., Monti, S., Shyu, C.-T. and Liu, C.-S.: Transition between the Okinawa trough backarc extension and the Taiwan collision: new insights on the southernmost Ryukyu subduction zone, Mar. Geophys. Res., 18(2), 163-187, doi: https://doi.org/10.1007/BF00286076, 1996.

Hsu, S.-K., Liu, C.-S., Shyu, C.-T., Liu, S.-Y., Sibuet, J.-C., Lallemand, S., Wang, C. and Reed, D.: New gravity and magnetic anomaly maps in the Taiwan-Luzon region and their preliminary interpretation, Terr. Atmos. Ocean. Sci., 9(3), 509-532, doi: 10.3319/TAO.1998.9.3.509(TAICRUST), 1998.

4) I did not understand why the high-resolution IIA (Figure 3) is shown for the Jiateng target only. What about Tainan?

Reply: According to the descriptions of historical reports and the previous study of Li et al. (2015), a local submarine mass failure could also be responsible for the Jiateng Harbor Flooding.   We then performed 378 unit-source tsunamis scenarios, where high-resolution bathymetry is required in order to better describe the regional submarine features, to form the near-field IIA.   On the other hand, the descriptions of 1782 Tsunami Event are different in the severity that the ocean waves "covered the whole of the submerged island so that no part of the flooded island remains visible except for the foot of the mountains" (Jäger, 1784).   Referring to the historical report above, it seemed that the 1782 Tsunami Event was affecting the whole island of Formosa instead of being limited to the local area of Tainan.   Therefore, we applied IIA method in larger areas (the southwestern coast and the far-field one).

Changes made:

No change was made.

5) What the sentence "the asperity effect was also applied" (line 155) means in practice? At least a brief sentence about this should be provided.

Reply: We thank the reviewer sincerely for the comment.   According to Somerville et al. (1999), "An asperity is a region on the fault rupture surface that has large slip relative to the average slip on the fault".   Wu (2012) mentioned that "the asperity is not directly connected to the tsunami earthquake. However, the tsunami height will be enlarged, if the asperity is located in an offshore area or on the tsunami ray".   In order to assess the worst-case tsunami hazard of Taiwan, Wu (2012) applied then the asperity effect to some of the tsunami sources in his study, including T02, the one we selected and used in this study.

Changes made (line 198):

For the worst-case tsunami hazard assessment, Wu (2012) also applied the asperities, which are regions on the fault that have larger slips compared to the average slip on the rupture area (Somerville et al., 1999) in some of the tsunami sources in his study.

References added:

Somerville, P., Irikura, K., Graves, R., Sawada, S., Wald, D., Abrahamson, N., Iwasaki, Y., Kagawa, T., Smith, N. and Kowada, A.: Characterizing crustal earthquake slip models for the prediction of strong ground motion. Seismol. Res. Lett., 70(1), 59-80, doi: 10.1785/gssrl.70.1.59, 1999.

6) At line 170, it seems that nonlinear equation were used for SMF only, but I guess that they have been used for the seismic sources as well. Authors should better clarify this point. Also, what is the reason for the selected Manning coefficient value (line 171)? 0.013 is a quite low friction, often used in presence of plain surfaces and no obstacles.

Reply: We thank the reviewer for the comment and apologize sincerely for missing this key information for the seismic tsunami sources.   The nonlinear shallow water equations were indeed used for the seismic tsunami scenarios as well.   The clarification has been added.

Indeed, just like the reviewer points out, 0.013 is a Manning coefficient value relatively low.   We are using this value as a conservative estimate since we don't really know what happened in the historical events, yet it seems to us a bit reckless not to apply any friction or higher friction values for the tsunami simulations.   Also, most of the southwest coast of Taiwan is fine-sand beaches.   Finally, the locations of Jiateng Harbor and Tainan are close to the lagoons which exist already in the Qing Dynasty.   According to the reasons above, we conservatively selected the Manning coefficient value 0.013 for the numerical simulations of this part.

Changes made (line 214):

To model the evolution of tsunamis, nonlinear shallow water equations were used with a bottom friction of Manning coefficient valued 0.013 for SMFs and seismic scenarios.

7) At line 183 is reported: "1 zhang approximately equals to 3 - 1/3 m" but I am not sure to have understood which is the exact correspondence in meters.

 Reply: We thank the reviewer for raising this question.   Actually, the length unit mentioned here confused us as well, though the original historical report is written in

Mandarin (Li et al., 2015).    Therefore, we suggest here using the villagers' houses and the bamboos as the height reference for estimating the tsunami height impacting the Jiateing Harbor.

Since "the rafters of the thatched roofs were all gone" (Li et al., 2015), and most of the residents at that time lived in one-floor houses, the tsunami height was estimated to be higher than the houses of the villagers.    Then according to Yu (1994) and the English translation of the supplementary element of Li et al. (2015), people were climbing upward (in the text, we know people climbed up the bamboos) as the water level rises.    Supposing that the reaction time and the climbing-up time were probably very limited for the residents, and after the flooding there was "One strong man jumped to ground, and helped others getting down" (Li et al., 2015), with the height that villagers could climb up within the limited reaction time and the height that one can jump down safely, the possible tsunami height estimated from the description is about two meters or higher in this study.

Changes made:

No change was made.

8) Section 3.1: I suggest to provide some more details on the position of the numerical gauge, as the water depth of the point and a zoomed map with the position respect with the coastline (an inset in Figure 5 could be enough)

 Reply: We acknowledge the reviewer for the suggestion.    Considering that the gauge locations are mentioned already in section 2.1, we added two zoom-in maps with the information (i.e. location and water depth) of the numerical gauge to the current Figure 1 since it is equally important to show more information about both the numerical gauges of the Jiateng Harbor and Tainan.

Changes made (Page 2):

[Figure]

Figure 1 The area of study. (a) Bathymetry and related structures off South Taiwan with 250 m contour interval. (b, c) Zoomed maps of Tainan and Pingtung (i.e. Jiateng Harbor location). The contours are plotted with 20 m and 50 m, respectively. Red triangles denote the numerical gauges in this study and the gauge information are marked in the right corner of each figure.

9) At line 235 I suggest to change the sentence in "[...] COMCOT seems to confirm [...]".

I would be more prudent on what the modelling results are indicating, since the parameters of the potential sources are quite uncertain for these events of the past. Modelling a few scenarios can help to support some hypotheses in broad terms, but I would be careful in drawing definitive conclusions.

Reply: We thank the reviewer for the suggestion. We should be more careful while making the conclusion.

Changes made (line 279):

The results of numerical simulations performed with COMCOT seem to indicate that the 1781 Jiateng Harbor Flooding and the 1782 Tsunami are two different events.

**Minor suggestions:**

line 12: "[...] , titled Taiwan Interview Catalague, [...]"

Reply: We thank the reviewer for the comment. The sentence in the abstract has been corrected.

Changes made (line 13):

The 1781 Jiateng Harbor Flooding, recorded by the Chinese historical document, titled Taiwan Interview Catalogue, took place on the southwest coast of Taiwan.

line 70-74: I suggest to slightly rephrase these sentences, since it seems to me that the reader could be a bit confused. I suggest something similar to the following: "Second, from the words "Voici ce que je lis dans J. L. Ab Indagine L. M." (i.e., "Here is what I read from J. L. Ab Indagine L. M.") in Perrey (1862), it is reasonable to believe that most of the content in that document is quoted from the German report Philosophisch und physikalische Abhandlungen (Jäger, 1784). After examining these two documents, it is also suspected that the date of a second letter sent from Beijing to

Versailles was reported with a typographical error about the year. Indeed, the date "En décembre 1682" is found in Perrey's [...] "

Reply: We agree with the reviewer and sincerely acknowledge the reviewer for the helpful suggestion.    The sentences have been rephrased accordingly.

Changes made (line 77):

Second, from the words "Voici ce que je lis dans J. L. Ab Indagine L. M." (i.e. "Here's what I read from J. L. Ab Indagine L. M.") in Perrey (1862), it is reasonable to believe that most of the content in the document is quoted from the German report Philosophisch und physikalische Abhandlungen (Jäger, 1784).    After examining these two documents.    It is also suspected that the date of a second letter sent from Beijing to Versailles was reported with a typographical error about the year.    Indeed, the date "En décembre 1682" is found in Perrey's (1862) record while the phrase is "Im Decemb. Des 1782" in Jäger's (1784) report, making the former less reliable.

line 106: "[...] where the submarine structures are located, [...]" ?

Reply: We thank the reviewer for the comment.    The sentence has been modified as the reviewer suggested.

Changes made (line 141):

The areas where submarine structures are located, […]

line 120: "broader"?

Reply: We apologize for the typo.    The sentence has been corrected accordingly.

Changes made (line 158):

Nevertheless, with a broader region in […]

line 167: I suggest to use quotation marks for the sentence ("subterranean movements causing the whole island to shake and be devastated; the earthquake lasted for 8 hours"), in order to emphasize the quotation.

 Reply: We acknowledge the reviewer for the suggestion. The quotation marks have been added to the sentence for emphasizing the quotation.

Changes made (line 210):

[…] the simulation was conducted based on the descriptions of the foreign reports of Jäger (1784) and Perrey (1862) ("subterranean movements causing the whole island to shake and be devastated; the earthquake lasted for 8 hours") and its location at the southern tip of Taiwan (Lin et al., 2009).

line 242: If I understand the sentence, I would rephrase as "Moreover, it is unlikely that a severe tsunami in 1782 was not reported by any Chinese document."

 Reply: We thank the reviewer for rephrasing the sentence, which makes the sentence much more understandable.    The sentence has been rephrased accordingly.

Changes made (line 286):

Moreover, it is unlikely that a severe tsunami in 1782 was not reported by any Chinese document.

line 244: "[...] the French record as reported by Perry (1862) [...]"

 Reply: We acknowledge the reviewer for the correction.    As the reviewer suggested, the sentence has been rephrased.    To avoid repeating the word "reported", we used the word "documented" instead.

Changes made (line 288):

Li et al. (2015) were the first group who read, translated and reported the French record as documented by Perrey (1862).

line 245: "[...] the historical documents cited by Perry [...]"

 Reply: We thank the reviewer for the correction.    Following the reviewer's suggestion, the sentence has been rephrased

Changes made (line 288):

[revised manuscript text omitted]

Dear reviewer,

We sincerely acknowledge the reviewer for the careful reading, helpful comments, and constructive advice. The manuscript has been revised according to the reviewer's suggestions in order to improve the quality. Responses to each comment could be found in the following part. The reviewer's comments are addressed in blueprints while we use the black and red prints to respond and mark the parts modified. The number of lines for the changing part is referred to the second version of the manuscript. All relevant changes are listed again at the end of this response file in line order.

Major comments:

1. The introduction section can be better structured. Instead of listing the historical records and previous studies, some scientific questions/current issues could be raised in the very beginning. In the end of the introduction, readers may expect a brief explanation of the content in the following sections.

Reply: We acknowledge the reviewer for the constructive suggestion. The beginning part of the introduction has been modified and the explanation of the content of each section has been added.

Changes made (line 24):

One of the major hazards in coastal regions is inundation by water waves generated by different mechanisms such as storm surges, tsunamis, or meteotsunamis. Storm surges and meteotsunamis are known to be triggered by weather events associated with pressure changes. On the other hand, tsunamis could be generated by earthquakes below or near the ocean, submarine landslides, volcanic eruptions, meteorite impacts, or off-shore rock falls, causing damage to the infrastructures and large loss of life in the coastal areas (Ghobarah et al., 2006; Mori et al., 2013; Widiyanto et al., 2019). On 15th January 2022, the volcanic eruption of Hunga Tonga-Hunga Haʻapai generated

tsunami waves that triggered evacuations in the surrounding countries. The sea-surface fluctuations were widely recorded and raised world-wild attention on the issue of coastal inundations (Carvajal et al.,2022; Manneela and Kumar, 2022; Ramírez-Herrera et al., 2022). To prepare for possible natural disasters, Li et al. (2018) have assessed the future tsunami hazard evolving with a rising sea level. Potential tsunami hazard assessments in the South China Sea (SCS) have also been extensively studied over recent years (Li et al., 2016; Liu et al., 2009; Megawati et al., 2009; Okal et al., 2011). However, it is still essential to look back and examine the historical events in order to recognize the regions that could be affected again. In fact, although most historical tsunamis were reviewed already (Lau et al., 2010; Terry et al., 2017), some of the events in Taiwan remain unknown.

[…]

(line 98):

In this paper, we assess the tsunami hazard of southwest Taiwan by a new analysis method and conducted numerical simulations to study the 1781 Jiateng Harbor Flooding and the 1782 Tsunami Event. The Impact Intensity Analysis (IIA) method and the modeling approaches of the tsunami scenarios are introduced in Sect. 2. Numerical simulation results are analyzed in Sect. 3 and the discussions through the literature records and the model results are presented in Sect. 4.

The following references were added:

Carvajal, M., Sepúlveda, I., Gubler, A., and Garreaud, R.: Worldwide signature of the 2022 Tonga volcanic tsunami, Geophys. Res. Lett., 49, e2022GL098153, Doi: 10.1029/2022GL098153, 2022.

Manneela, S. and Kumar, S.: Overview of the Hunga Tonga-Hunga Ha'apai Volcanic Eruption and Tsunami, J. Geol. Soc. India 98, 299–304, doi: 10.1007/s12594-022-1980-7, 2022.

Ramírez-Herrera, M.T., Coca, O., and Vargas-Espinosa, V.: Tsunami Effects on the Coast of Mexico by the Hunga Tonga-Hunga Ha'apai Volcano Eruption, Tonga, Pure Appl. Geophys., doi: 10.1007/s00024-022-03017-9, 2022.

2. The authors mentioned the possibility of storm surges could not be ruled out, one paragraph reviewing the most serious impact posed by past storm surge events could be very helpful for readers to understand the potential magnitudes.

Reply: We thank the reviewer sincerely for the suggestion. Helping the readers to have a picture of how catastrophic that storm surge events could be is important. Following the reviewer's suggestion, we added a brief review of the storm surge cases.

Changes made (line 83):

Severe coastal impacts caused by storm surges have been recorded and studied. The Bay of Bengal was struck by Bhola cyclone in 1970 and more than 250,000 human lives were taken away resulting from the storm surge and the flooding it triggered (Hossain et al., 2008). In 2005, the storm surge associated with Hurricane Katrina damaged tremendously the Gulf of Mexico, peaking at over seven meters along the Mississippi coastline (Fritz et al., 2007; Robertson et al., 2007). The high-water marks of Cyclone Nargis surpassed the 2004 Indian Ocean Tsunami run-up height at corresponding locations in May 2008 (Fritz et al., 2009), causing over 100,000 deaths in Myanmar. In southwest Taiwan, a unique local ritual (i.e. Quianshuizang) is still being practiced today to pacify the victims of the 1845 Yunlin Kouhu Storm Surge Event (Chang, 2019). The storm surge, inducing by a super-typhoon, was responsible for the fatalities of over 2,000 (Tsai, 2014; Yin, 2013), and was followed by famine and plague (Chang, 2019). [….]

The following references were added:

Chang, C.-J.: Documenting life in the era of climate change: Huang Hsin-yao's Nimbus and Taivalu. Asian Cinema, 30(2), 235-254, doi: 10.1386/ac_00006_1, 2019.

Fritz, H. M., Blount, C., Sokoloski, R., Singleton, J., Fuggle, A., McAdoo, B. G., Moore, A., Grass, C., and Tate, B.: Hurricane Katrina storm surge distribution and field observations on the Mississippi Barrier Islands, Est. Coast. Shelf Sci., 74, 12–20, doi: 10.1016/j.ecss.2007.03.015, 2007.

Fritz, H. M., Blount, C., Thwin, S., Thu, M., and Chan, N.: Cyclone Nargis storm surge in Myanmar, Nat. Geosci., 2, 448–449, doi: 10.1038/ngeo558, 2009.

Hossain, M. Z., Islam, M.T., Sakai, T., and Ishida, M.: Impact of Tropical Cyclones on Rural Infrastructures in Bangladesh, Agric. Eng. Int. CIGR E J. Invited Overview, 2: 1-13. ISSN:1682-1130, 2008

Robertson, I., Riggs, R. Yim, S., and Young, Y.: Lessons from Hurricane Katrina storm surge on bridges and buildings, J. Waterw. Port, Coast. Ocean Eng. 133(6), 463–483, doi: 10.1061/(ASCE)0733-950X(2007)133:6(463), 2007.

3. After reading the newly added historical records in German, the possibility of volcanic eruption cause should be more carefully reevaluated, especially after the Tonga volcanic tsunami which occurred in Jan 15, 2022.

Reply: We agree with the reviewer.   The two major active volcanoes of Taiwan actually locate in the north.   Chen et al. (2001) presented a thermoluminescence (TL) method and found that the volcanic eruptions of the turtle island (Kueishantao) occurred later than 7 ka.   On the other hand, the Tatun volcano group has been proved to be active although there have been no eruption records in human history (Belousov et al., 2010; Pu et al., 2020).   We addressed the mud volcanoes, which located in the offshore area of southwest Taiwan in our discussion part carefully in the new revision.

Belousov, A., Belousova, M., Chen, C.-H.and Zellmer, G. F.: Deposits, character and timing of recent eruptions and gravitational collapses in Tatun Volcanic Group, Northern Taiwan: hazard-related issues. J. Volcanol. Geotherm. Res. 191, 205–221, doi: 10.1016/j.jvolgeores.2010.02.001, 2010.
Chen, Y.-G., Wu, W.-S., Liu, T.-K., and Chen, C.-H.: A date for volcanic eruption inferred from siltstone xenolith, Quat. Sci. Rev., 20:869–873, doi: 10.1016/S0277-3791(00)00047-0, 2001.
Pu, H.-C., Lin, C.-H., Lai, Y.-C., Shih, M.-H., Chang, L.-C., Lee, H- F., Hong, G.-T., Li, Y.-H. Chang, W-Y.and L, C.-H.: Active Volcanism Revealed from a Seismicity Conduit in the Long-resting Tatun Volcano Group of Northern Taiwan, Sci. Rep., 10(1), 6153, doi: 10.1038/s41598-020-63270-7, 2020.

Changes made (line 298):

[…] Meanwhile, since active submarine mud volcanoes were observed in the offshore area of southwest Taiwan (Chen et al., 2014; Mai et al., 2021), we shall neither rule out

the possibility of multiple hazards occurring coincidently or very close in the same period (e.g. The Eruption of Mount Pinatubo and Typhoon Yunya in 1991).    Therefore, numerical simulations of storm surges or complex hazards should be taken into consideration in future studies.

The following references were added:

Chen, S.-C., Hsu, S.-K., Wang, Y., Chung, S.-H., Chen, P.-C., Tsai, C.-H., Lie, C.-S., Lin, H.-S., and Lee, Y.-W.: Distribution and characters of the mud diapirs and mud volcanoes off southwest Taiwan. J. Asian Earth Sci., 92, 201–214, doi: 10.1016/j.jseaes.2013.10.009, 2014.

Mai, H.-A., Lee, J.-C. Chen, K.-H., and Wen, K.-L.: Coulomb stress changes triggering surface pop-up during the 2016 Mw 6.4 Meinong earthquake with implications for earthquake-induced mud diapiring in SW Taiwan, J. Asian Earth Sci., 108, 104847, doi: 10.1016/j.jseaes.2021.104847, 2021.

Minor comments:

1.  Line 32, a more detailed historical records should be added in the reference, Lau et al., 2010, NHESS, Written records of historical tsunamis in the northeastern South China Sea – challenges associated with developing a new integrated database.

Reply: We acknowledge the reviewer for sharing and reminding the importance of the relevant literature.    The research of Lau et al. (2010) has been for historical tsunami records.

Changes made (line 35):

[revised manuscript text omitted]

---

## Author Response (AR2)

Responses to the Editor's Corrections

Dear Editor,

We sincerely acknowledge the editor's careful reading and the detailed corrections, which certainly helped us improve the manuscript's quality. The manuscript has been revised following the annotated version of this paper from the editor. The revisions are shown in balloons in the 'track changes' file. In addition to correcting the typos and rephrasing the sentence, we also made the following modifications as the editor required and suggested:

Figure 6:

As the editor required, we marked the position of the numerical gauge in the maximum wave height figures with the black cross. To be clear, we also addressed the numerical gauge location of the Jiateng Harbor again in the figure caption (shown already in Fig.1 and line 126). The changes to the figure caption are marked in red prints.

[Figure]

Figure 6 (a) Maximum wave height of the 1781 SMF tsunami simulations. The black cross denotes the numerical gauge of Jiateng Harbor (120.467°E, 22.417°N). (b) Numerical gauge records at the Jiateng Harbor study site for the SMFs scenarios of FR, FS and KC. One-meter elevation was marked with dashed lines for comparison.

Text A1:

The original French text is available indeed. It has been provided in Text A1 before the translation part as the editor suggested. The changes are marked in red prints.

Text A1

Original text from Gazette de France.: De Paris, le 12 Août 1783.

Une lettre de la Chine fait mention d'un évènement arrivé l'année dernière, & peut-être plus terrible encore que ceux qu'ont éprouvés la Sicile & la Calabre dans le commencement de celle-ci féconde en désastres. En attendant une relation plus détaillée, voici ce que l'on en raconte : Le 22 Mai de l'année dernière, la mer s'éleva sur les côtes de Fo-Kien à une hauteur prodigieuse, & couvrit presqu'entièrement pendant huit heures l'île de Formose qui en est à 30 lieues. Les eaux, en se retirant, n'ont laissé à la place de la plupart des habitations que des amas de décombres sous lesquels une partie de la population immense de cette Isle est restée ensevelie. L'Empereur de la Chine, voulant juger par lui-même des effets de ce désastre, est sorti de sa capitale ; en parcourant ses provinces, les cris de son Peuple excités par vexations de quelques Mandarins, ont frappé ses oreilles ; & on dit qu'il en a fait justice en faisant couper plus de 300 têtes.

(Translated from French to English by T.-C. Liu, June 2020)

A letter from China mentions an event that happened last year, and perhaps it's more terrible than those experienced in Sicily and Calabria, where suffers disasters a lot in the beginning. Pending a more detailed narrative, here's what they tell us: On May 22, 1782, the sea level rose to a prodigious height on the coast of Fujian, covering almost the entire island of Formosa for 8 hours, which situated 30 league (1) away. The water ebbed, leaving most of the residential areas nothing but piles of rubble, under which a part of the huge population stayed buried. Wanting to inspect the effects of this disaster by himself, The Emperor of China left the capital. During the journey going through his provinces, the cries of his People, whipped up by the irritation of some Mandarins, struck his ears, and it is said that he executed over 300 people.

(1) The ancient measurement unit "lieue" is about 4 km long in length.

Line 441:

We added the address of Gazette de France, which could be found in Gallica, the digital library of the National Library of France in the reference.

Gazette De France.: De Paris, le 12 Août 1783," in Gazette De France, Organe officiel du Gouvernement Royal, Paris, pp. 288 (in French), https://gallica.bnf.fr/ark:/12148/bpt6k6254403x/f4.item, 1783.